# Green Roof Development in ASEAN Countries: The Challenges and Perspectives

Hanny Chandra Pratama [1] [ID], Theerawat Sinsiri [1,2,*] and Aphai Chapirom [2]

1    School of Civil Engineering, Suranaree University of Technology, Nakhon Ratchasima 30000, Thailand;
hannychandra.arch@gmail.com

2    Sustainable Innovation and Energy-Efficient Construction Materials (SIE-CON),
Suranaree University of Technology, Nakhon Ratchasima 30000, Thailand; aphai_ch@g.sut.ac.th

*    Correspondence: sinsiri@sut.ac.th; Tel.: +66-803347718

**Abstract:** Green roofs (GRs) have emerged as an essential component for the sustainability of buildings, as they reduce the need for cooling energy by limiting heat transmission into building space. The benefits of implementing GRs are appropriate in tropical regions with hot temperatures. The entire Association of Southeast Asian Nations (ASEAN) is located in a tropical climate and receives about 12 h of sunlight every day throughout the year, which offers excellent opportunities to install GRs. This research reviews the literature on GR knowledge in ASEAN countries over the past decade (2012–2022) and discusses two main points including (i) GR development level status and (ii) GR performance regarding drivers, motivations, and barriers. The review reveals that Singapore and Malaysia are two among ten countries with significant developments in GRs. Barriers to expertise, government regulations, and public awareness of green roofs represent the most challenging aspects of GR implementation in ASEAN countries. Although research regarding the use of green roofs has been conducted widely, ASEAN countries still need to investigate regulatory breakthroughs, incentives, and technology applications to encourage the use of GRs. The review recommends promoting the use of GRs, which have the potential to reduce energy consumption by up to fifty percent, outdoor surface temperature up to 23.8 °C, and room temperature to 14 °C. The use of GRs can also mitigate runoff issues by up to 98.8% to avoid the risk of flooding in ASEAN countries, which have high rainfall. In addition, this review sheds new insights on providing future potential research to improve GR development in the ASEAN region.

**Keywords:** green roofs; tropical green roofs; cities; sustainable development

## 1. Introduction

Changes in a city's appearance caused by population density have a substantial effect on land-use change, replacing green spaces and vacant land with urban infrastructure [1]. Among all the continents in the world, Asia has the biggest population, which accounts for around 61% of the global population (4.7 billion). The world's most populated country is China (1438 million), followed by India (1380 million), the United States (331 million), Indonesia (273 million), and Pakistan (220 million), and four of the top five most populous countries are located in mainland Asia [2]. Population density can lead to numerous issues, such as: greenhouse gas emissions, climate change, the urban heat island (UHI) effect, air and water pollution, and frequent flooding, owing to the lack of green spaces, which have necessitated the development of innovative responses to the issues of urban living [3–5].

Green (vegetated, eco, or living) roofs are basically roofs planted with vegetation on top of a growth medium (substrate). The concept was designed and developed to be incorporated into urban infrastructure to offset the effects of climate change, urban expansion issues, and other potential difficulties related to human intervention in ecosystems. GRs generally comprise several components, including vegetation, substrate, filter fabric,

drainage material, root barrier, and insulation. Each component plays a mutually beneficial role in the system that forms the benefits of GRs [6,7].

GRs can enhance the energy efficiency of buildings and help mitigate the UHI effect by lowering the ambient temperature and improving thermal comfort for humans [8,9]. Additionally, GRs help to store rainwater and delay peak flow, reducing the risk of floods. Some of the rainwater will be absorbed by the substrate or trapped in the pore spaces. It can also be absorbed by plants and retained in plant tissues or transpired back into the atmosphere [10]. GRs buffer acidic rain and theoretically retain pollutants, thereby producing good quality stormwater runoff [10,11]. Sound absorption is another feature of GRs; sound can be minimized by absorbing sound waves diffracting over roofs [12]. The GR system is a popular approach that could help to mitigate air pollution in urban environments. More precisely, on a sunny day, GRs may lower the $CO_2$ concentration in the nearby region [13,14]. GRs also aid in restoring biodiversity that has been lost due to urban development and offer a safe place for birds, insects, and different plants to grow [15].

GRs can be classified by their purposes and characteristics into two major types: intensive roofs and extensive roofs. Intensive green roofs (IGRs) need a deep substrate and require skilled labor, irrigation, and constant maintenance. IGRs are usually associated with roof gardens and need additional structural support due to the heavy weight of the substrate [16,17], while extensive green roofs (EGRs) have a relatively thin layer of soil and grow sedums or moss. EGRs require lower maintenance and cost less, resulting in more widespread application compared to IGRs [18–20].

As GRs have become more popular in the recent decades and their implementation has expanded beyond Europe, the urge to understand how this novel ecosystem functions has arisen in many areas. Various studies have been conducted to assess the novelty and suitability of green roofs in non-tropical areas, but only few have been conducted in tropical areas [21]. The tropical region contains a greater amount of daylight than does the temperate zone. For this reason, an evaluation is needed to measure how to adapt the design to the climatic features of tropical green roofs.

In recent years, there have been many publications on GRs. According to Web of Science, more than 2400 articles, reviews, book chapters, and proceedings papers on green roofs have been published in the last 30 years (1989–2018). Individual continents show that most publications were issued in Europe (37.84%), followed by Asia (29.11%), North America (24.93%), Australia (4.72%), and South America (2.49%), with the lowest number of publications in Africa (0.91%) [22]. In Asia, a closer examination shows that research has come from only several countries, including China, Japan, South Korea, India, Taiwan, and Iran. Even though ASEAN countries dominate the tropical region in Asia, relatively little green roof literature has been published. Out of approximately 10 ASEAN member states, including Brunei, Cambodia, Indonesia, Laos, Malaysia, Myanmar, the Philippines, Singapore, Thailand, and Vietnam, only Singapore has made significant green roof research contributions [23].

Therefore, this paper aims to identify the current conditions and perspectives related to GRs in ASEAN countries by reviewing the research articles based on ASEAN cases. The paper then addresses the GR development level in ASEAN countries as well as discusses the key findings observed in the drivers, motivations, and barriers of GRs. We anticipate the study will provide qualified support to urban planners and policymakers for urgent future research to advance the development of GRs in ASEAN countries.

## 2. Materials and Methods

This study uses a systematic literature review (SLR) to eliminate bias and increase scientific value. According to Uman [24], systematic literature review examines the findings of previous studies in order to recognize recurring and consistent patterns. The three pillars of rigor, transparency, and replicability form the basis of a systematic literature review (SLR), which allows deeper understanding of a subject and identification of knowledge

gaps [25]. The SLR adopted in this article follows the PRISMA protocol according to Moher et al. [26], and the number of stages of the PRISMA protocol varies according to the authors.

For this study, a comprehensive literature search was undertaken to identify papers specifically published between 2012 and 2023. To search for the articles, we utilized Scopus, MDPI, Science Direct, and industry reports. There were ten search country terms within the title, abstract, and keywords of the articles on the significance of green roofs in ASEAN countries, including ("green roofs") AND ("Brunei" OR "Cambodia" OR "Indonesia" OR "Laos" OR "Malaysia" OR "Myanmar" OR "Philippines" OR "Singapore" OR "Thailand" OR "Vietnam"). Inclusion and exclusion criteria include language, timeline, and literature type.

For data analysis, the title, abstract, keywords, journal name, authors' names, and year of publication of the searched records were exported to an MS Excel spreadsheet. Two independent authors screened the paper titles and abstracts independently. Then, two independent reviewers performed an eligible assessment by carefully screening the full text of papers that were screened in the first screening process to assess their fit with the inclusion criteria. During this stage, disagreements between reviewers were resolved through arbitration by the other two reviewers.

Data that were exported to an MS Excel spreadsheet were modified by adding items for which data were sought for data management. As shown in Figure 1, the PRISMA approach includes two rounds of elimination. The first round of screening classifies the articles into conceptual, theoretical, and empirical studies. The second screening is an evaluation that involves reading the full texts and classifying them into one of six GR development level categories according to Zhang's methods [27]. From the review of 99 total full-text articles, we selected 70 articles, which were included for qualitative synthesis review. Out of these 70 full-text articles, with publication dates ranging between 2012 and 2022, 47 articles (67%) were qualitative, and 23 articles (33%) were quantitative.

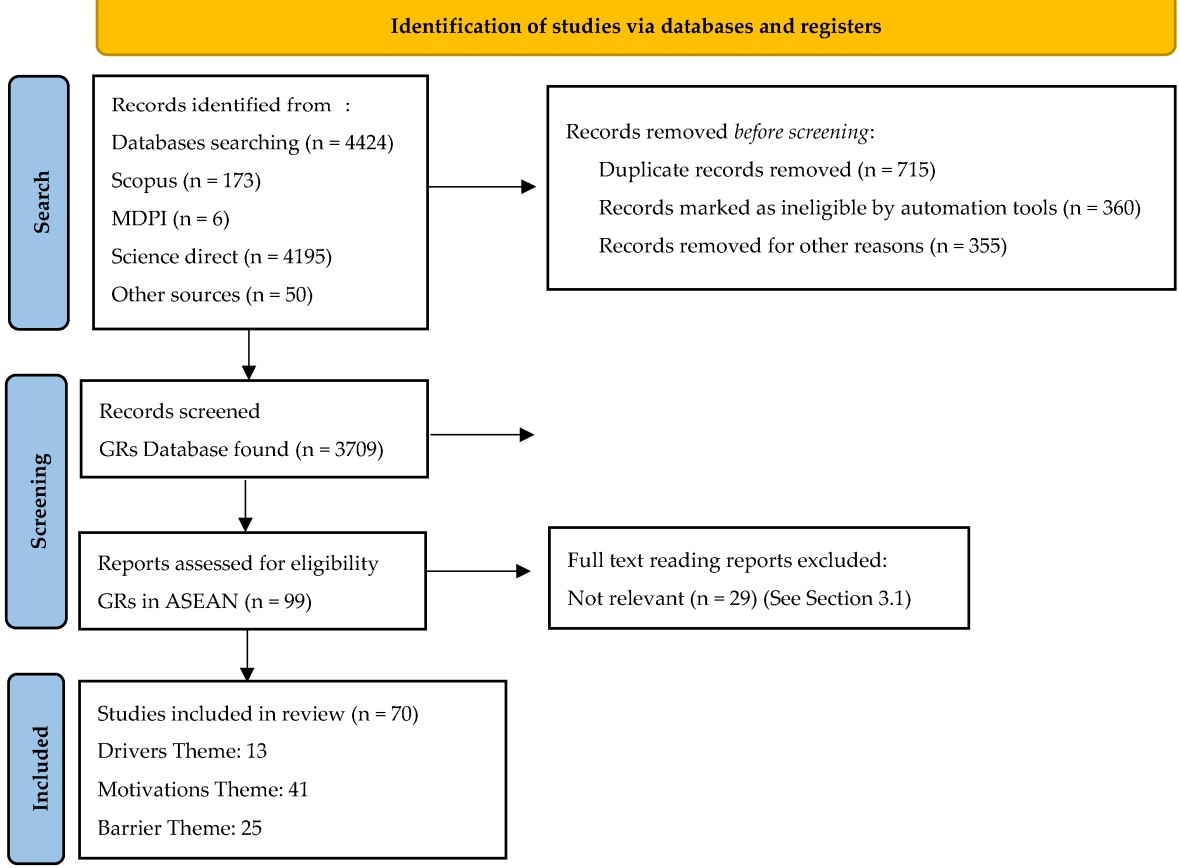

**Figure 1.** PRISMA flow diagram for green roof database in ASEAN countries.

Selected literature refers to the number of studies providing evidence, despite the fact that clearly, not all studies provide evidence of equal quality. We extrude all the result data in selected literature by systematically presenting the group theme of research (drivers, motivations, and barriers) to determine the scope and gaps in the ASEAN countries' study of GRs. We anticipate the study will provide qualified support and can help researchers design urgent GR future research in ASEAN countries.

## 3. Results

### 3.1. Mapping of Selected Literatures

The academic journals gathered using searching strategies resulted in a total of 99 papers.

The filtering process outlined in the previous section selected the 70 most relevant studies for a full-text review. Studies that did not meet the eligibility criteria comprised reviews, studies comparing green roofs with other technologies but not nongreen roofs, and studies presenting no comparisons. Such studies were not included in the selected literature.

Figure 2 provides the details of the papers obtained initially, regarding the number of papers selected for the review based on the ASEAN country cases. It is shown that Malaysia has a significant amount of literature available on the contribution of GRs as compared to other countries for the GR drivers; this includes studies reporting policy pressure from government as encouragement to use GRs (e.g., Ismail et al. [28], Fauzi et al. [29,30]) and advance technology of GRs (e.g., Munir et al. [31], Mahdiyar et al. [32]). When it comes to the motivations behind using GRs, several studies have provided examples, including those related to the energy efficiency of GRs (e.g., Sittipong and Pichai [33]); the urban heat island (UHI) effect, such as studies that report the benefits of GRs to outdoor surfaces (e.g., Munir et al. [34]); and indoor room temperature (e.g., Irsyad et al. [35]) as compared to conventional roofs. Moreover, motivation to use GRs is also derived from their capacity to control runoff, including storm water retention (e.g., Orozco et al. [36]) and runoff water quality (e.g., Romali et al. [37]). There are also data regarding the motivation for using GRs to increase biodiversity (e.g., Sananunsakul [38]). Lastly, in investigating barriers of GRs in ASEAN countries, a large number of studies addressed the problem. These could be grouped into three categories: government policy and technological barriers, which were only found in Singapore and Malaysia; economic barriers, as when installation costs are highly volatile (e.g., high price GRs in Brunei [39] and low price GRs in Vietnam [40]); social acceptability and feasibility barriers, which are discussed in numerous studies.

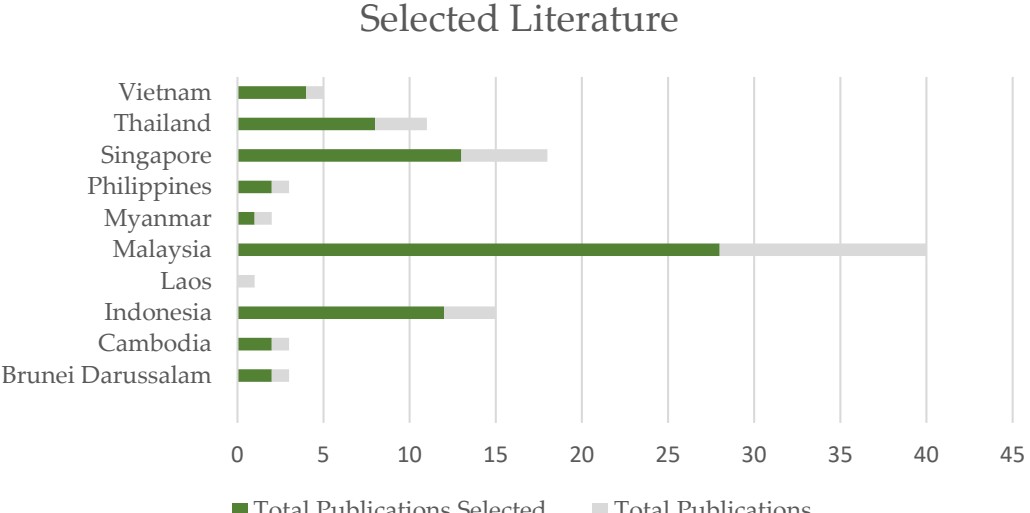

**Figure 2.** Schematic representing total search findings and the number of studies sorted for full-text review.

Figure 3 shows that the majority of studies conducted in ASEAN countries focused on the motivations for implementing GRs, totaling 52%. The motivations are still the focus since ASEAN societies are still attempting to validate the advantages of using GRs and overcome the main barriers. On the other hand, driver research is still insufficient. Drivers are urgently needed to help ASEAN countries accelerate the development of GRs. The findings of each selection criteria will be described in the stages of green roofs in ASEAN countries (Section 3.2). In addition, the reviews of green roofs' potential in ASEAN countries as well as the social acceptability and feasibility issues will be provided (Section 3.3).

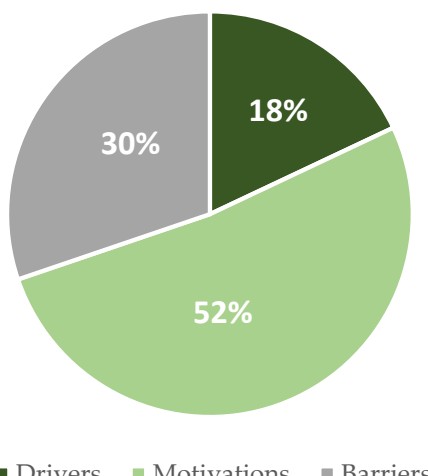

■ Drivers    ■ Motivations    ■ Barriers

**Figure 3.** Schematic representing total type of selected literature, including drivers, motivations, and barriers.

### 3.2. Stages of Green Roofs in ASEAN Countries

The existence of green buildings as approaches to environmental restoration has increased in urban green spaces during the past decade to reduce climate change. Figure 4 illustrates the number of urban green spaces in various major Asian cities.

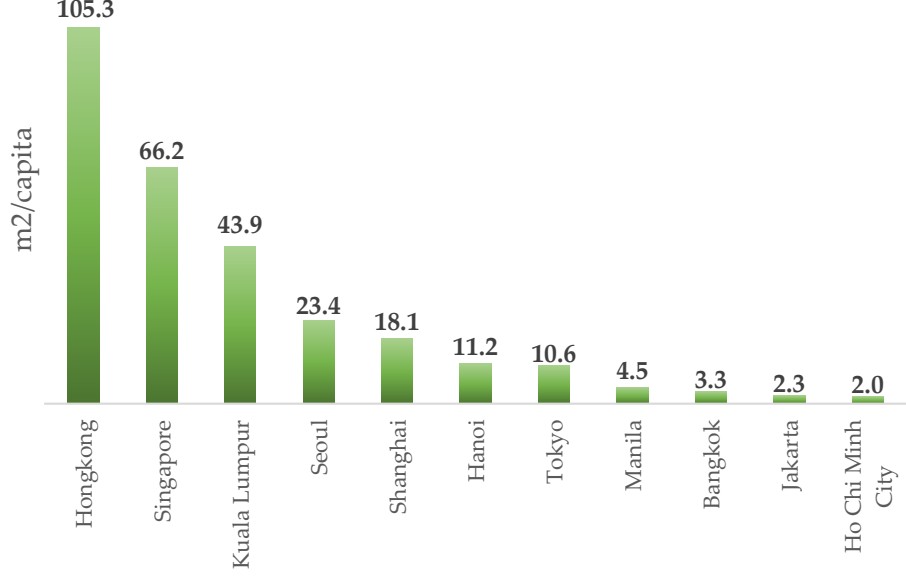

**Figure 4.** Number of urban green spaces in several countries in Asia. (This figure is illustrated using the data from Hasnan, available online in theaseanpost.com [41]).

In the data, there are several metropolises that are included in ASEAN countries, Singapore (66.2 m$^2$/capita); Kuala Lumpur, Malaysia (43.9 m$^2$/capita); Hanoi, Vietnam (11.2 m$^2$/capita); Manila, Phillipines (4.5 m$^2$/capita); Bangkok, Thailand (3.3 m$^2$/capita); Jakarta, Indonesia (2.3 m$^2$/capita); and Ho Chi Minh City, Vietnam (2.0 m$^2$/capita) [41]. However, WHO suggested a standard of 9 square meters per capita. Moreover, a previous study mentioned the ideal urban green space value of a modern city is 50 m$^2$ per capita. [34,42]. Singapore, Malaysia, and Vietnam are the three leading ASEAN nations that have met the minimum standards for green space. In addition, Singapore has exceeded the ideal indicator of urban green space, making Singapore a modern ASEAN city. As observed, Singapore's approach is similar to Hong Kong's, which includes the implementation of a three-dimensional green space system. This is demonstrated by the existence of numerous green projects, including Gardens by the Bay, Singapore Botanic Garden, and NUT Green Roof. Malaysia, on the other hand, has to add approximately 7 m$^2$/capita to resemble Singapore as a modern city. The implementation of an ideal green area could increase the city's wealth and property values. This is also related to the fact that GRs can boost people's happiness and lower the risk of depression [43,44]. In addition, according to UNICEF, the creation of urban green areas can considerably enhance the physical, mental, and social development of children [45].

GR technology has developed as a solution for combating the lack of urban green space. However, GR growth in developing countries is slower than in developed nations. This is also influenced by the number of GR research publications in developed countries, with USA, China, and Italy dominating the contributions with total percentages of around 22%, 13%, and 9% while developing countries (India and Iran) are below 2%. Interestingly, Singapore, a developed country in the ASEAN region, cannot contribute much to green roof research, as it only contributes under 2% [22]. Lack of interest, applicable policies, and standards, as well as incompetence, are the most frequently cited obstacles to knowledge and awareness in developing countries [46].

GR implementation is a complicated issue concerning many aspects. According to Zhang [27], GR implementations are started from "Environmental and Social Problems" that plague cities in Stage 1 (e.g., heatwaves, urban warming, and urban flooding). The problems resulting from "Diverse Pressure", such as in environmental pressure (e.g., energy, water, and air), social pressure (e.g., health, safety, mortality, and morbidity), economic pressure (e.g., expenditure, productivity, and market) and policy pressure (e.g., urban sustainability, energy saving, and water saving) are included in Stage 2. The pressure from various aspects drives a response for tackling the problem through "GR Research and Development" in Stage 3, where some studies have been conducted to monitor, model, and assess GR performance.

After the preliminary approach from Stages 1 to 3, GRs have actually been implemented in the "Promotion" Stage 4, where regulation and stimulation have been started to support further steps in GR development. In "GR Implementation" Stage 5, a systematic process is relevant to practical GR application in cities, including design, construction, contract and supply, operation, and maintenance. "Performance Evaluation" Stage 6 is evaluation in terms of GRs' function in addressing environmental and social issues in cities. Figure 5 shows the framework of green roof development. Furthermore, several studies or actual projects in ASEAN countries will be briefly evaluated and classified according to their stages of development below.

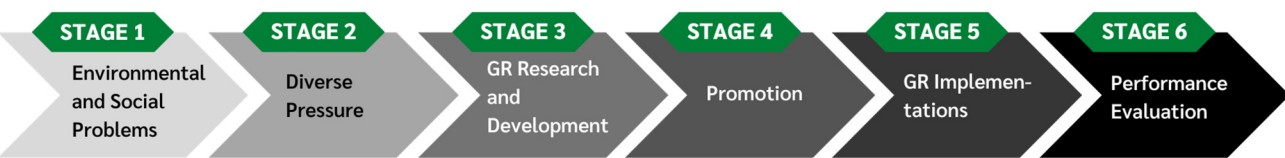

**Figure 5.** The framework for green roof development, promotion, and application. (This figure is adapted from the previous study by Zhang [27]).

Brunei is also known as Brunei Darussalam. The country, which is one of the relatively small countries in the ASEAN region, is still considered to be in "Diverse Pressure" (Stage 2) as a result of several challenges in developing GRs. GR development in Brunei started from Stage 1, where increasing temperature increases electrical consumption. The government is interested in implementing GRs in an effort to reduce energy consumption by an estimated 30 percent [47]. This energy issue necessitates Stage 2 to evaluate public interest in GRs. According to results of an actual survey in Bandar Sri Bengawan, the capital city of Brunei, only 7% of the 414 buildings constructed using flat-roof reinforced concrete have the potential for GR installation. People prefer pitched roofs because of their strength and prevention of leaks. Moreover, GRs' appeal to society is as of yet minimal. A survey of building owners' interest in the usage of GRs revealed that only 24% of respondents supported the implementation of GRs [48] The level of public awareness of GRs could be increased by educating buildings' owners and increasing the involvement of professional knowledge and government sectors.

For the past decade, Cambodia has experienced flood problems in the capital city of Phnom Penh. This is due to limited capacity of the city's drainage system. Lyna mentioned that several areas in Phnom Penh were affected by flooding with a height of more than one meter [48]. The current issue in Cambodia is classified as Stage 1, which motivates researchers to solve the flood problem by improving green infrastructure. Tree canopies, bioswales, permeable pavements, and green roofs are some of the green infrastructure options proposed by Nou in his study [49]. In the development of green infrastructure, the use of GRs can be suggested according to previous research by Sarkar indicating that GRs can reduce runoff control by 18 to 29% [50].

Indonesia is located in the equatorial zone with high average temperatures during daytime and relative exposure time. The high temperatures in Indonesia appear to be a major issue in Stage 1 for Indonesia. Jakarta, as the metropolitan city in Indonesia, has average daytime temperatures that could reach up to 35 °C. The surface of a roof exposed to the sun during the day can be affected by temperatures as high as 50 °C [51]. In order to limit the heat absorption into buildings, the development of GRs in Indonesia is important. An evaluation study in Bandung has undertaken Stage 2 to analyze the deployment of GRs in the targeted area. GRs can be applied on the slab concrete roofs of 53 out of 222 dwellings. Sloped roofs that may not be suitable for GRs installation are still prevalent in Indonesia [52]. Several GR research studies and developments, which are categorized as Stage 3, are also being conducted. A study by Munir found that the use of lightweight foamed concrete and a green roof reduces construction loads and cooling loads for air conditioning systems [51]. Moreover, as an alternative to concrete, corrugated zinc roofing could be used as the base of a building's green roof. Yuliani mentioned that corrugated zinc roofs have the fastest response to cold down to 2 °C compared to concrete-based GRs [53]. Unfortunately, there is no research for GR-supporting regulations in Indonesia.

Malaysia is one of the ASEAN member countries with positive GR development. It accounted for 28 papers in the research database on GRs, the highest of all ASEAN countries. Based on a survey by Rahman, the 30 cases where projects integrated GRs categorized it as Stage 5. Interestingly, GRs are popularly implemented in condominiums and offices with 11 projects and 5 projects, respectively [53]. However, mostly, GR projects in Malaysia are privately accessed only for those who own condominiums or work at offices. It may be an obstacle to promote GRs on small scales such as with residential housing, schools, or low-rise buildings. However, some research projects in Malaysia have reached Stage 6, and performance evaluations of the implementation of GRs on public buildings have been reviewed by Rahmat [54]. He mentioned that GRs might reduce the average surface temperature by between 3.6 °C and 11.1 °C compared to conventional roofs. Evaluations of the performance of actual buildings are expected to present to the market the fact that GRs increase building performance.

Myanmar's government initiated implementation of smart city projects in Yangon and Mandalay. However, references and standards for green buildings in Myanmar remain

inadequate. Lwin mentioned that "energy efficiency" and "water efficiency" are the most crucial problems, categorized as Stage 1 [55]. Lwin also mentioned the necessity of further development of a green building rating system for Myanmar, which should be considered along with the domestic conditions of the country and categorized as Stage 2 [55]. In addition, there is still no specific information on GR development in Myanmar. GRs can reduce building electricity consumption and improve water quality through absorption, making them a viable option for future research in Myanmar to increase energy efficiency and water efficiency.

Urbanization from rural to urban areas in the Philippines has resulted in 51.2% of the population moving and residing in cities, categorized as Stage 1 [56]. The development of Stage 3 GRs has also been carried out through research to increase water retention. The results show that adding 0.6% hydrogel to the soil can increase water retention to 45.46%, which is higher than the 30.07% achieved by GRs without hydrogel [57]. GRs have been implemented in the Philippines, and it is evident that there are several GR suppliers in the Philippine construction market, which is classified as Stage 5. Orozco surveyed a number of local GR suppliers in the Philippines. The results revealed that green roofs generally met minimum standards [36]. By adjusting GR products based on environmental and climatic conditions of the Philippines, providing local materials, and developing GR experts, it is possible to overcome the aforementioned problems. However, a study about "Promotion" Stage 4 from the government has not been found.

Currently, Singapore is a thriving, world-class urban center with limited land and resources. However, Singapore's high level of greenery is largely the result of decades of careful planning. The Singapore Sustainable Development Blueprint maximizes land use while maintaining 8 m$^2$ of green space per person and increasing the amount of greenery in high-rise buildings to 50 ha by 2030 [23]. Another government institution, Singapore National Parks (NParks), has introduced a GR incentive to encourage owners of existing buildings to have GRs, which is categorized as Stage 4. The program supported approximately 50% of GR installation costs in order to promote GR application and create a green urban environment [58]. Referring to the installation of GRs as categorized as Stage 5, Singapore has remarkable GR projects that have attracted many building owners in Singapore. Projects such as Nanyang Technological University (NTU) have been known all around the world as stunning works of architecture (See Figure 6). NTU received the Green Mark Platinum Award from the Singapore Building and Construction Authority (BCA) for adopting best practices in environmental sustainability. The GR performance evaluation, which is categorized as Stage 6, showed that the GRs in NTU help with energy savings of nearly 120,000 kWh per year and water savings of over 1170 m$^3$ per year, resulting in lower operation and maintenance costs. The green roof, high-performance glass, and carbon dioxide sensors in its air-handling units also contribute to the building's energy efficiency and healthy indoor air quality [59]. From the references above, it shows that government and architects have played a significant role in promoting GRs, particularly from a design perspective. The integrated development of GRs, from Stage 1 of "Environmental and Social Problems" with land availability to Stage 6 of "Performance Evaluation" with the large green roof project at NTU, demonstrates Singapore's position as the ASEAN leader in the development of green roofs.

Moving to Thailand, one of the big countries in the ASEAN region, the major obstacles to adoption are the absence of adequate subsidy programs and deficient knowledge and skilled labor force; these typical ASEAN pressures are categorized as Stage 2. The interview study of Sangkakol [60] mentioned that the roof is the last part of a building to be completed—typically after the majority of available funds have been allocated. Therefore, the absence of GR subsidies make GRs unlikely to be installed, and GRs might be skipped even if planned initially. Interestingly, in Thailand, as shown in Figure 6, Thammasat University (TU) built a magnificent structure known as the largest urban rooftop farm in Asia, which we categorized as Stage 5. The 22,000-square-meter TU GR combats climate change by combining modern landscape architecture with traditional agricultural ingenuity,

urban farming, a solar roof, and green public space [61]. However, research articles on GRs in Thailand are still limited. The expansion of research and promotion in Thailand has the potential to increase development of GRs for all segments of society.

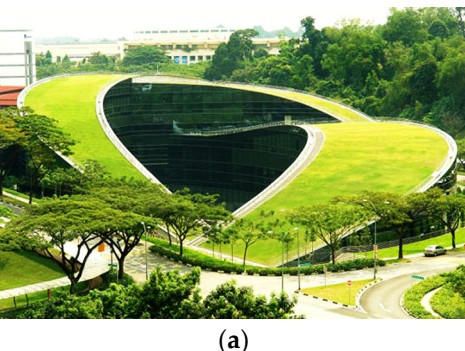
(**a**)

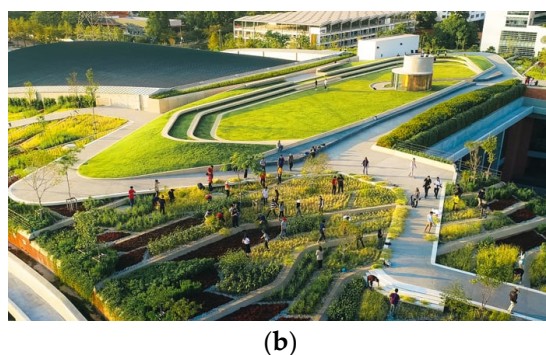
(**b**)

**Figure 6.** The magnificent green roof projects in ASEAN countries: (**a**) Nanyang Technological University (NTU), Singapore and (**b**) urban rooftop farm, Thammasat University (TU), Thailand.

Several major Vietnamese cities, including Hanoi and Ho Chi Minh City, have minimal green space areas. Hanoi has only 11.2 m$^2$/capita of urban green space, while Ho Chi Minh City has only 2.0 m$^2$/capita [41]. Pam's responder poll, which is an evaluation of "Diverse Pressure" (Stage 2), received a positive response, with 64 percent of building users and 62 percent of specialists agreeing to promote green space [62]. This response is far better than the poll in Brunei, where just 24% of building users participated. A study that can be classified as Stage 3 showed that the price of green roof construction in Vietnam is approximately 15 to 45 USD/m$^2$, which is fairly inexpensive compared to other green roof pioneering nations such as Hong Kong at around 400 to 1000 USD/m$^2$ [63] or Singapore at approximately 89 to 197 USD/m$^2$ [63]. It is well demonstrated that Vietnam's potential for green roofs is highly promising [40]. As a recommendation, it is necessary to conduct additional studies on Stages 4 to 6 in light of Vietnam's vast potential for GR development.

Identifying research on GRs conducted in ASEAN countries, shown in Figure 7, allowed us to pinpoint the critical stages of GR development level. In general, 6 of 10 ASEAN countries are still restricted under Stage 3 "GR Research and Development". However, only Singapore is stable in all aspects of highest stage development. Malaysia, who is on an equal level of development as Singapore, is still attempting to tackle deficiencies, such as the "Promotion" aspect of Stage 4, where government regulations are still limited (shown in Table 1). Subsidies for constructing GRs as a development strategy in Singapore are an effective strategy. However, GR subsidy programs are not implemented in Malaysia. Moving on to the Philippines and Thailand, these two nations have adopted GRs up to Stage 5, but they face the same issues: a lack of regulatory support from the government, subsidies, end-user desire, and green roof expertise. Indonesia and Vietnam are still in Stage 3 of green roof development and research. Implementation remains low in Indonesia and Vietnam. Even though Indonesia's green roof research database is ranked third, research on green roof technological improvements remains limited. This results in stagnation of GR development at Stage 3. Therefore, research on the promotion, implementation, and performance evaluation of GRs is urgently required in order to expand the use of GRs. Brunei, Cambodia, Laos, and Myanmar are lagging behind the other ASEAN nations in the development of GRs. Particularly, in Laos it is still quite difficult to find previously published studies on GRs. It is recommended that those four countries conduct more research on GR benefits to obtain potential data on GR efficiency in specified locations in the countries.

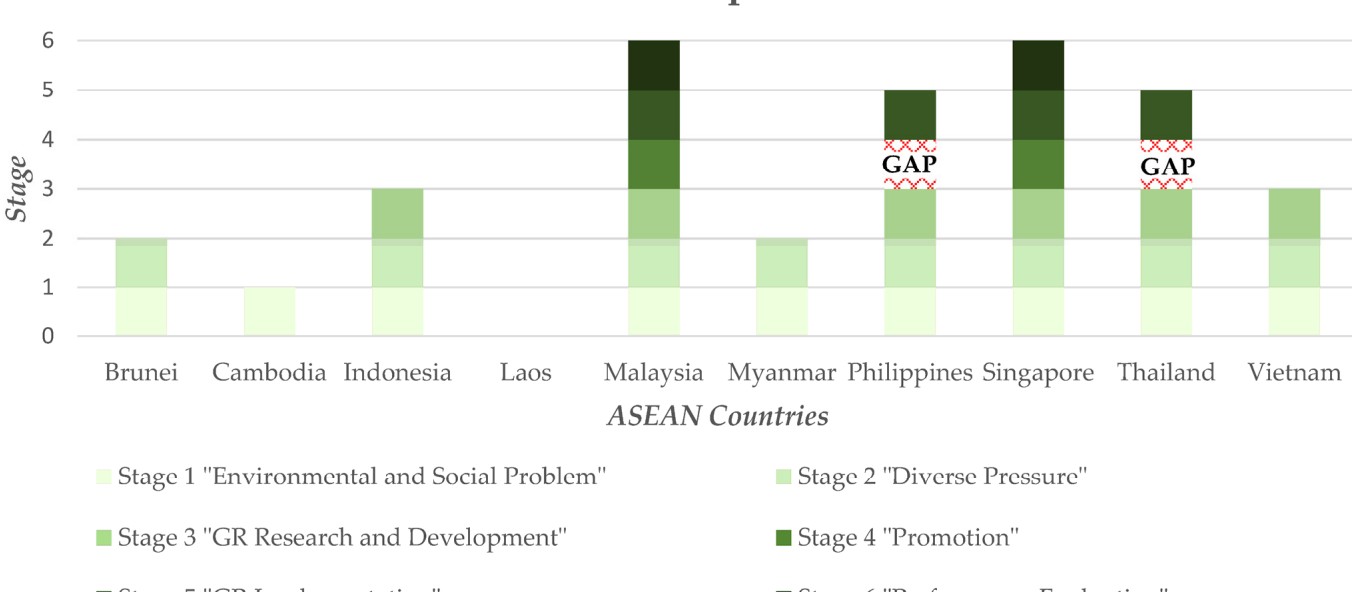

**Figure 7.** The green roof development levels in ASEAN Countries.

### 3.3. Green Roof Performance in ASEAN Countries

In this study, green roof performance is categorized into three review groups: drivers, motivations, and barriers research. In total, 70 research studies are included in this section.

3.3.1. Drivers of Green Roofs in ASEAN Countries

In this section, we report the results of "5 articles on Policy Pressure" and "7 articles on Advance Innovation Technology", which discuss drivers of GRs in ASEAN countries. The factors below are crucial and can accelerate the development of green roofs in the ASEAN region:

- Policy Pressure.

To fully implement GRs, mandatory or voluntary policies, regulations, guidance, standards, or initiatives are necessarily established. Some examples of developed cities and countries that encourage the development of GRs are Toronto and Canada, who implement policy pressures that aim to legislate sustainable construction [64].

Regulations and standards have a significant role in the development of GRs in Singapore and Malaysia because of the existence of a handbook in Singapore and maintenance guidelines in Malaysia, as shown in Table 1.

**Table 1.** Policy pressure to drive GR implementation, as reported in the literature.

| Authors | Key Findings | | Country |
|---|---|---|---|
| | **Subsidy Scheme** | **Regulations and Standards** | |
| Ismail et al. [28] | The scheme supported 50% of green roof installation costs by Singapore National Parks (NParks) to encourage owners to have green roofs | Standards for Singapore GRs,<br>• Introduction to Vertical Greenery<br>• Handbook of Skyrise Greening in Singapore<br>• Vertical Greenery for the Tropics | Singapore |

**Table 1.** *Cont.*

| Authors | Key Findings | | Country |
| | Subsidy Scheme | Regulations and Standards | |
| --- | --- | --- | --- |
| Fauzi et al. [29,30], Siew et al. [65], Zaid et al. [66] | None | • Applying GR will improve the green building assessment standards up to 7–10%<br>• BREEM code is suitably used for assessment.<br>• Evaluation of roof penetration barrier, drainage, filter, substrate, vegetation, green building credit, and thermal performance guides.<br>9 guidelines for GR maintenance | Malaysia |

However, this appears to be inextricable from the subsidy scheme. In Singapore, the subsidy scheme has been implemented for quite some time, making building owners more interested in installing GRs. However, there is no report found that Malaysia implemented this subsidy scheme. Abdul Rahim [67] also reports that the lack of subsidies in Malaysia is one of the obstacles to enormous propagation of GRs in society. The study concludes that government has a significant impact on the development of green roofs. Government action in the form of subsidies is a highly successful way of encouraging building proprietors to install green roofs.

• Advance Innovation Technology.

GR technology's first appearance was in Ziggurat of Ancient Mesopotamia, from the fourth millennium until 600 BC. The GRs were located in the courtyard temples, and shrubs and trees were planted on terraces formed by a gran-stepped pyramid [64]. GRs have become more successful over time as a result of technical developments. Environmental considerations influence the development of GR technology in every country; thus, there are always variations in the application of technology between nations. Therefore, technological innovation is one of the drivers encouraging the advancement of GR development.

Table 2 shows the advanced innovation technology literature that has been researched in ASEAN countries. There are notable differences in the technological development of GRs. The countries in Stage 6 development focus on developing smart and efficient green roof construction technology. Mahdiyar's research on prototypes established which style of GR was most suitable for a building [32]. In addition, Singapore also has artificial technology as one of its advanced technologies [68]. Meanwhile, in other countries, green roof research is still concentrated on adapting GR implementation to the country's culture. In Indonesia and Thailand, for instance, alternative materials are developed for GR optimization in cost and performance [31,69]. However, the development of advanced technology for GRs in ASEAN countries is still limited compared to the total GR articles in the ASEAN region.

**Table 2.** Advanced innovation technology to drive GR implementation, as reported in the literature.

| Authors | Findings | Details |
| --- | --- | --- |
| Mahdiyar et al. [32] | Development of a decision support system (DSS) for selecting the optimum type of GR for residential buildings in Kuala Lumpur | Malaysia, All types GR, System to selection GRs |
| He et al. [68] | Artificial neural network (ANN) models had a better average prediction for accurate modelling of thermal and hydrological performance | Singapore, EGR, Evaluate plant evapotranspiration |
| He et al. [70] | A methodology for a quick evaluation of thermal performance of a green roof during the early stages of design | Singapore, EGR, Model development of RTTV |

**Table 2.** *Cont.*

| Authors | Findings | Details |
|---|---|---|
| Phoomirat et al. [33] | Develop a rapid assessment checklist (RAC) to assess GR services | Thailand, All GRs, Assessment tools |
| Munir et al. [31] | The panel 400 mm width is an optimal form for the structural roof deck of the green roof system | Indonesia, EGR, Structural deck precast foamed concrete for GRs |
| Shahid et al. [71] | Palm oil clinker has good ability to drain excess water, and there is no side effect in terms of plant development | Malaysia, EGR, Palm oil clinker as drainage layer |
| Chandra et al. [69] | Lightweight cellular concrete 1200 and 1400 kg/m$^3$ meets the standard to substitute normal concrete | Thailand, EGR, Structural deck alternative for GRs |

### 3.3.2. Motivations for Green Roofs in ASEAN Countries

In this section, we report the results of total 41 research articles including eight studies in "Energy Efficiency", 15 articles in "Urban Heat Island Mitigation", 16 articles in "Runoff Control", and two articles in "Biodiversity Increase". The factors below are important and can validate the benefits of applying GRs in ASEAN countries:

- Energy Efficiency.

In the five studies reporting on annual energy consumption, the effect of green roofs ranged from savings of 1–57% (Table 3). Installing GRs in tropical countries effects a in reduction heat gain and leads to decrease in cooling loads. Energy savings is affected by a number of factors, including layer thickness, moisture content, and vegetation selection [72]. Sittipong [73] reported that an increase in the thickness of the soil layer reduces the building energy consumption. As shown in Figure 8, two different thickness layers were compared by installing multiple sensors within a GR's layers. Wong [74] discovered that shrubs are recommended because they effectively reduce the energy consumption of buildings. The findings also agreed that an increase in soil thickness would further lower the building's energy consumption. In addition, the moisture content of the soil, which is influenced by vegetation, can have a significant impact on the outcome. Apart from this, in the case of multi-story structures, green roofs reduce energy consumption significantly on the floors directly below the green roof. Azis [75] collected data of electrical usage on the top floor and lower floor. The results show that the top floor was directly impacted by the green roof and consumed 40% less electricity.

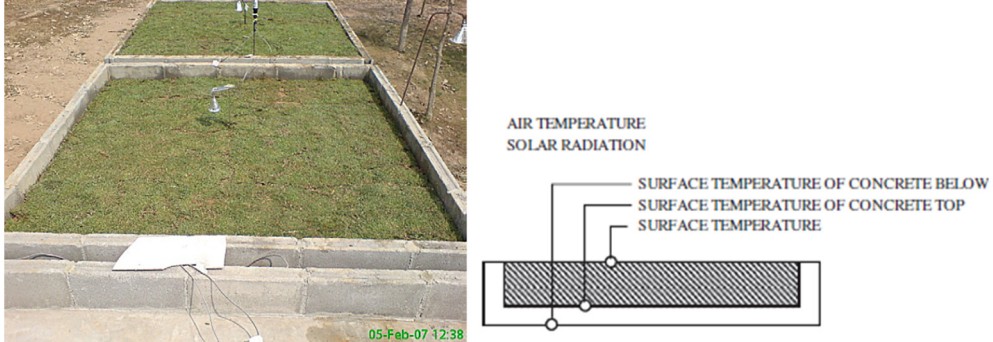

**Figure 8.** Experimental set of GR model in Thailand [73]. "Reproduced with permission from Sittipong Permpituck, Pichai Namprakai, The energy consumption performance of roof lawn gardens in Thailand; published by Elsevier, 2012".

**Table 3.** Energy savings due to GR implementation, as per the selected literature.

| Authors | Energy Savings | Details |
|---|---|---|
| Wong et al. [74] | 1–15% reduction in energy consumption | Singapore, EGR, Comparison rooftop garden with turfing, shrubs, and tree |
| Dewi et al. [76] | 11.24–21% reduction in cooling load | Indonesia, EGR, GR simulations by planting *Tradescantia Spathacea* and Sedum acre |
| Sittipong and Pichai [73] | GR thickness 0.10 m–31.07% GR thickness 0.20 m–37.11% | Thailand, EGR, Increasing soil depth improves energy savings by around 6% |
| Azis et al. [75] | 37–40% reduction in energy consumption | Johor Bahru, Malaysia, EGR and IGR GR integration with 17-floor apartment |
| Yuliani et al. [77] | Concrete GRs, 57.1% reduction in heat flow Corrugated Zinc GRs, 50.8% reduction in heat flow | Surakarta, Indonesia, EGR, A corrugated zinc roof with plants could improve thermal conditions in a building |

- Urban Heat Island Mitigation.

Green roofs are regarded as one of the most practical tools in mitigating the UHI effect because they reduce air and surface temperatures in cities through the provision of shading, removal of heat from the air, and reduction of temperatures of the roof surface. In this section, we report the results of 15 studies and discuss the impact of GRs on the UHI effect. Outdoor surface and indoor room temperature in green roof studies will be discussed:

1. Outside Surface Temperature.

The maximum reduction ranges for outside surface temperature fell between 10.4 and 23.8 °C compared to conventional roofs (Table 4) according to the 11 studies including actual survey, experimental, and simulation studies. The results of an actual survey on constructed GRs are lower than those of an experimental study that was merely a trial model. However, the reduction performance of GRs in ASEAN countries is highly effective. Green roofs are more effective in reducing temperature during warm temperatures. ASEAN countries, located in tropical climate areas, have consistent temperature radiation throughout the year. Moreover, six out of ten ASEAN countries have reached a maximum temperature above 40 °C [78]. Therefore, the application of green roofs in the ASEAN region is highly recommended.

Aside from climate, material selection has an influence on decreasing performance due to temperature. In Indonesia, Munir [51] revealed that lightweight foamed concrete in combination with roof slabs for green roofs has positive results on cooling as well as on construction loads, which will also impact cost effectiveness. Based on a simulation study, another alternative from Yuliana [53] claimed that corrugated zinc with plants can be used instead of green roof concrete.

**Table 4.** Reduction of outdoor surface temperature after implementing GRs, as reported in the selected studies.

| Authors | Max. Outdoor Surface Temperature Reduction | Ambient Temperature | Remarks |
|---|---|---|---|
| Sunakorn [79] | 10.4 °C | 30 °C | Thailand, Actual Survey Study, EGR, Local material Thailand (Green Mat) can be applied on GRs |

**Table 4.** *Cont.*

| Authors | Max. Outdoor Surface Temperature Reduction | Ambient Temperature | Remarks |
|---|---|---|---|
| Qin [80], Yang et al. [81] | 19 °C | 24–32 °C | Singapore, Simulation study, Cool roofs, Insulation layer on cool roofs plays a negligible role in heat flux values |
| Munir et al. [51] | 19.1 °C | 28–57 °C | Aceh, Indonesia, Experimental study, EGR, Precast foamed concrete, Combination of lightweight foamed concrete and GR |
| Ahmed and Rumana [82], Ismail et al. [83] | 19.8 °C | 30.2 °C | Malaysia, Actual survey study, Roof garden, GRs performed better than white concrete roof and gardening roof |
| Yuliani et al. [53,77] | 20 °C | 24–35 °C | Surakarta, Indonesia, Experimental study, EGR, Concrete and corrugated zinc, A corrugated zinc roof with plants could improve thermal conditions in a building |
| Azis et al. [75], Rahmat et al. [54], Rahman et al. [84] | 23.8 °C | 30–36 °C | Kuala Lumpur and Putrajaya, Malaysia, Experimental study, EGR, Green roof improving public environment |
| Yuliani et al. [53,77] | 20 °C | 24–35 °C | Surakarta, Indonesia, Experimental study, EGR, Concrete and corrugated zinc, A corrugated zinc roof with plants could improve thermal conditions in a building |
| Azis et al. [75], Rahmat et al. [54], Rahman et al. [84] | 23.8 °C | 30–36 °C | Kuala Lumpur and Putrajaya, Malaysia, Experimental study, EGR, Green roof improving public environment |

2. Indoor Room Temperature.

Indoor room temperature has an effect on human thermal comfort. According to Ahmed [82], the temperature comfort zone for humans is between 23.5 °C and 28.5 °C; thus, if the room temperature is within this range, humans can be present without the need for heating or cooling. Implementing GRs reduces the thermal transfer in building, which will also impact indoor room temperature reductions.

The maximum reduction ranges for indoor temperature are from 3 to 14 °C (Table 5) according to nine studies, including actual survey, experimental, and simulation studies. According to several studies [35,82,85], the application of a roof garden has the same favorable effects as the usage of GRs. Combining the use of GRs and a roof garden will improve GR heat resistance.

The materials used also affect the performance of GRs at inside room temperatures. Munir [51] used lightweight foamed concrete for GRs with an effective outdoor decrease up to 10.2 °C. The use of GRs can result in significant energy savings since the temperature of a room that applies GRs deviates less from the comfort zone than the temperature of a room that does not apply GRs. This allows cooling to operate more efficiently in spaces with GRs.

**Table 5.** Reduction of indoor temperature after implementing GRs, as reported in the selected studies.

| Authors | Max. Indoor Temperature Reduction | Temperature Comparison | Remarks |
|---|---|---|---|
| Yuliani et al. [77] | 3 °C | GR 23–28 °C NON-GR 27–31 °C | Surakarta, Indonesia, Experimental study, EGR, Concrete and corrugated zinc, A corrugated zinc roof with plants could improve thermal conditions in a building |
| Pahm et al. [62] | 3 °C | GR 31 °C NON-GR 34 °C | Ho Chi Minh City, Vietnam, Actual survey study, roof garden In cooler area, GR still resulted lower than non-GR temperature |
| Ahmed and Rumana [82], Ismail et al. [83], Rahman et al. [84] | 3 °C | GR 24.5–31.5 °C NON-GR 25.5–33 °C | Malaysia, Actual survey study, roof garden, GRs performed better than white concrete roof and gardening roof |
| Irsyad et al. [35] | 6.8 °C | GR 23–26 °C NON-GR 24–32 °C | Bandung, Indonesia, Experimental study, EGR, The use of Amaranta and Bromelia plants can inhibit heat transfer into the room |
| Sunakorn [79] | 7 °C | GR 24.6–27.9 °C NON-GR 29.6–39.9 °C | Thailand, Actual survey study, EGR, Local material Thailand (Green Mat) can be applied on GRs |
| Munir et al. [51] | 10.2 °C | GR 35–38 °C NON-GR 42–51 °C | Aceh, Indonesia, Experimental study, EGR, Precast foamed concrete Combination of lightweight foamed concrete and green roof |
| Yang et al. [81] | 14 °C | GR 26–27 °C NON-GR 28–41 °C | Singapore, Simulation study, EGR, Insulation layer on cool roofs plays a negligible role in heat flux values |

- Runoff Control.

1. Storm Water Retention.

Retention is the process of holding water on the green roof and preventing water from draining off the green roof [86]. Table 6 presents the water retention capacities given in the selected literature in several ASEAN countries. The overview of studies reported that GRs in ASEAN countries can have a maximum water retention capacity up to 98.8%. The main parameters affecting the retention values are rainfall intensity and duration. Comparatively, subtropical regions in China receive approximately 1058 mm of precipitation annually, allowing them to retain up to 100 percent of precipitation [86]. In contrast, the Philippines' tropical rainfall of 2348 mm per year can retain up to 75 percent. In addition to rainfall, the ability of GRs to retain water is affected by factors such as sunshine intensity, temperature, soil media depth, and plant selection.

2. Runoff Water Quality.

Stormwater runoff carries non-point source pollution from cities, impacting water quality. GRs have the potential to reduce runoff quantity, while their impact on runoff quality is also considered. GRs are a source of pollutant reduction if the pollutants are fewer in the runoff than in the rainwater or in the runoff from bare roofs [11]. Figure 9 shows the example of runoff model based experiment in Singapore.

**Table 6.** Water retention capacity on GRs, as reported in the selected studies.

| Authors | Water Retention Capacity | Remarks and Influence Factors |
|---|---|---|
| Chanrachna et al. [49] | 37.90% | Phnom Penh, Cambodia, Comparison study, GI (Green Infrastructure) scenario by installation tree, bioswales, permeable pavements, and green roofs |
| Cuong et al. [85] | 42% | Hanoi, Vietnam, Experimental study, EGR, Four models with different irrigation systems |
| Asinas et al. [57] | 46% | Philippines, Experimental study, EGR, Waterproof substrate improving water retention by adding hydrogel to substrate media |
| Chai et al. [87] | 72.5% | Kuching, Malaysia, Simulation study, EGR, Green roof water balance by virtual modelling approach |
| Orozco et al. [36] | 75% | Metro Manila, Philippines, Comparison study, EGR, Suppliers company, Comparison of suppliers with six different GR technologies |
| Ayub et al. [88], Fai et al. [89], Chow et al. [90], Romali et al. [37], Kok et al. [91], Chai et al. [87], Musa et al. [92] | 86% | Malaysia, Simulation study, EGR, GRs decrease of stormwater runoff, Types of substrates resulting in different runoff, Mixture of plant species was the most effective vegetation at reducing runoff water |
| Vergroesen and Joshi [93] | 98.8% | Singapore, Experimental study, EGR, 3-day experiment, Different sedum comparison |

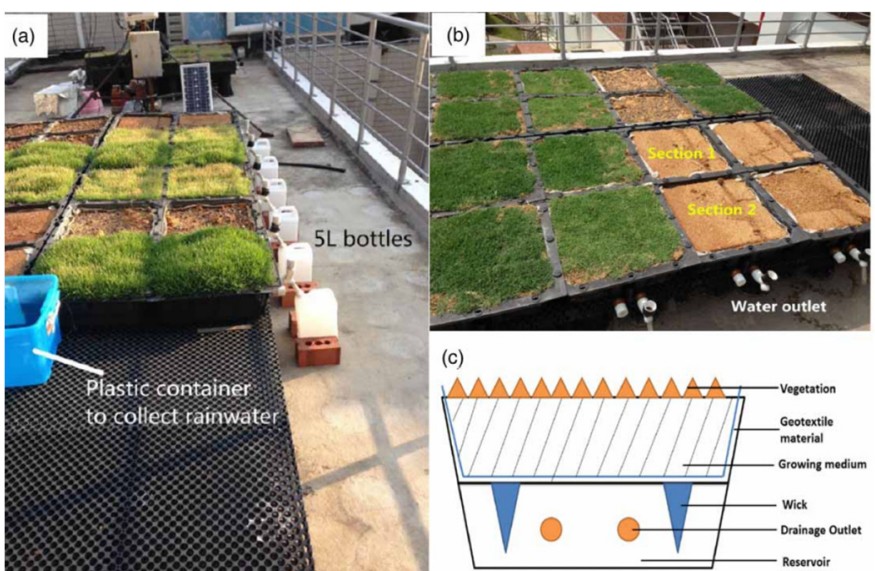

**Figure 9.** (**a**–**c**) Experimental set of runoff-model-based experiment in Singapore by Lim et al. [94].

In several studies in developed countries, GR runoff released higher concentrations of total phosphorus, total nitrogen, nitrates, and phosphates, meaning GRs are a source of these nutrients [95,96].

Studies on runoff quality are currently limited; Malaysia and Singapore have a few studies on this matter from GRs. Table 7 shows the runoff quality studies in Malaysia and Singapore. However, study of runoff quality in other ASEAN countries needs to be carried out to develop and validate the benefits of green roofs to improve water quality.

**Table 7.** Runoff quality on GRs, as reported in the selected studies.

| Authors | Findings | Remarks and Influence Factors |
|---|---|---|
| Romali et al. [37] | GRs with beach morning glory improve the COD up to 99% GRs with creeping oxeye reduce the BOD up to 17% | Malaysia, Experimental study, EGR, Green roof with beach morning glory and creeping oxeye |
| Lim et al. [94] | Mixture substrate clay (5–30%), silt (5–60%), and sand (20–75%) can decline in chemical concentration level | Singapore, Experimental study, EGR, Substrate comparison |
| Vijayaraghavan et al. [97] | The concentrations of most of the chemical components in runoff were highest during the beginning of rain events and subsided in the subsequent rain events | Singapore, Experimental study, EGR, Real rain events and ten different simulated rain events |

- Biodiversity Increase.

  Green roofs can promote urban biodiversity through the provision of complex vegetation structures, greater foraging and roosting options for animals, and enhanced habitat connectivity [94].

  An observation study of biodiversity of GRs in Singapore revealed that spontaneous species had approximately 28.9% site cover in a year apart from planting seeds at the beginning. In addition, after installing the GRs, the fauna increased up to 69 fauna species, including species of birds, bees, hornets, wasps, butterflies, moths, etc. [98]. Sananunsakul stated a similar perspective that green roof characteristics may influence bird communities utilizing these urban ecosystems [38].

3.3.3. Barriers to Green Roofs in ASEAN Countries

GR implementation is also constrained by barriers of government policy, technology barriers, economic issues, social acceptability, and feasibility issues. A search with the green roof database in ASEAN countries found that the study of barriers is dominant compared to other green roof topics. We report the results total 25 articles including five studies in "Economics" and 20 articles in "Social Acceptability and Feasibility Issues":

- Government Policy and Technology Barriers.

  As described above, regarding ASEAN drivers with green roofs, green roofs can be developed with governmental support and technological advancements. Singapore and Malaysia are the only two ASEAN countries supported by government regulations and rigorous programs. On the other hand, eight other ASEAN countries did not have any research or archives pertaining to government sponsorship and technology. This is the biggest barrier in ASEAN countries that must be addressed for the green roofs of ASEAN countries to advance.

- Economy.

  In this section, we discuss the findings from our review in terms of economic barriers. We shortlisted five studies that discussed the economic feasibility of GRs with the inclusion of several costs and benefits. Figure 10 presents the types of GRs and their costs in each of the studies. A study in developed countries showed that the installation cost for extensive green roofs can range between 55 and 220 USD/m$^2$ [99]. The high cost of installation is a major barrier to the development of GRs in ASEAN countries. This is demonstrated by

Rahman in Brunei Darussalam with typical green roof installations ranging from 208 to 318 USD/m$^2$. The roof slab's design for excessive strength has a significant impact on the high-cost green roof. As a recommendation, the structural load design that meets the load requirements of the green roof will be extremely cost-effective [39].

## Installation Cost per Meter

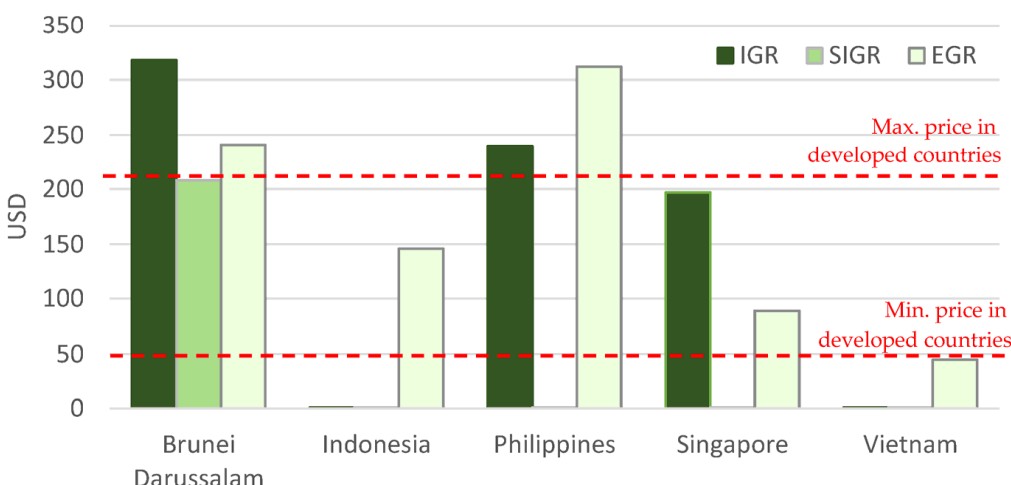

**Figure 10.** The green roof development level in ASEAN countries. (Source: Brunei Darussalam [39], Indonesia [100], Philippines [36], Singapore [101], and Vietnam [40].

More importantly, Vietnam's pricing range is significantly lower than the average price in industrialized markets. This may be because conventional green roofs are still manufactured in Vietnam [40]. The advancement of the application of green roof technology is urgently required to enhance the performance of green roofs while maintaining market-competitive pricing.

Research on the cost of green roofs in Singapore reveals that prices are comparable to the global average for green roofs. In addition, a complete green roof system and excellent performance back this claim [101]. The rise of green roof construction in ASEAN countries will be stimulated by the creation of high-quality green roofs at prices that are far lower than the world average.

● Social Acceptability and Feasibility Issues in ASEAN Countries.

We discovered around 12 studies examining the perspectives of possible GR implementers and users, such as architects, engineers, property developers, and building owners. Social acceptability from numerous studies shown in Table 8 can represent the major obstacle points. According to most studies, initial building expenses, maintenance costs, and lack of knowledge are the primary obstacles to the broad use of GRs. Moreover, the lack of implementation knowledge and qualified labor, as well as the absence of government incentives, are additional causes for concern.

According to the findings of Shams, survey research on user perception in Brunei resulted less interest in GRs [48]. Several studies in Malaysia indicated that the lack of GR expertise in the country remained a significant issue [67,102]. Arif [103] described that a method for enhancing public interest in green roofs can be initiated by introducing GRs to the community in public places. This will enhance the user's ability to experience the advantages of a green roof.

**Table 8.** Social acceptability according to GR implementers and users.

| Challenge | Brunei | Malaysia | Indonesia | Thailand | Vietnam | Myanmar | References |
|---|---|---|---|---|---|---|---|
| Lack of Interest using GRs | ✓ | ✓ | ✓ | | ✓ | | [48,52,104–107] |
| Lack of Guidelines | | ✓ | | | | ✓ | [67,102,108] |
| Less Government Intensive | | ✓ | | ✓ | | ✓ | [60,103,109] |
| High Installation Cost | | ✓ | | | | | [110] |
| Lack of Expertise | | | | ✓ | | | [105,106] |

Observing the absence of social acceptability in ASEAN countries also has an effect on building aspects. Table 9 describes research studies discussing the current building feasibility for applying GRs. Briefly, most limitations have been found in studies. Shams [48] mentioned that the application of GRs on structures in Brunei is quite limited. Luthfiyyah conveyed a similar result [52], in which the city demonstrates the absence of potential buildings for applying GRs. Additionally, Chow mentioned the warnings and concerns regarding use of GRs in industrial buildings.

**Table 9.** Current building feasibility for applying GRs.

| Authors | Findings | Remarks |
|---|---|---|
| Suparwoko and Taufani [111] | The potential to use GRs as an urban farming space | Sleman, Indonesia, Survey study, Feasibility of buildings |
| Chow and Abu Bakar [102] | Industries were very cautious about having rooftop gardens due to the unknown risk on maintenance aspects | Malaysia, Survey study, Feasibility of buildings |
| Rahman et al. [112] | GRs help to regenerate and revitalize the commercial setting in urbanized areas | Malaysia, Survey study, Feasibility of buildings |
| Hwang and Roscoe [113] | Green roofs as a platform for the interaction between humans and nature is a possibility that is yet to be fully explored, especially in a tropical context | Singapore, Actual study, Feasibility of buildings |
| Naing et al. [114] | 47% of selected buildings in sampling area were possible to retrofit for a green roof | Thailand, Survey study, Feasibility of buildings |
| Jalanugroh and Tikul [115] | Of twenty-six education buildings, three can be built with an intensive green roof, seven can be built but gardeners are needed, and sixteen can be built with extensive green roofs | Thailand, Survey study, Feasibility of buildings |
| Lwin and Panuwatwanich [108] | Energy efficiency, water efficiency, and waste and pollution, are the most crucial categories that should be concerned | Myanmar, Survey study, Feasibility of buildings |

According to our findings, ASEAN society is still unaware of GRs. However, if GRs have an affordable and visually acceptable design, they have the potential for social acceptability. Furthermore, user-specific experiences and expectations are essential to consider. To expand their awareness, there is a need to directly introduce the benefits of GRs to society.

## 4. Discussion

### 4.1. Challenges of GR Development in ASEAN Countries

The study presented here stems from the phenomena of uneven development of GRs in the ASEAN countries, even though the ASEAN region has potential topographical characteristics for maximizing the benefits of green roofs. This is evident from the skewed

distribution of the 70 GR research articles in ASEAN, Malaysia, and Singapore, which accounted for more than half of the distribution with a total of 41 articles. (There were 28 articles for Malaysia and 13 articles for Singapore). This results Malaysia and Singapore's acceleration toward an advanced level of GRs.

Based on the inclusion criteria for a paper in the review, the evidence for the drivers in ASEAN countries was limited to thirteen articles, which were dominated by Malaysia and Singapore. To increase the level of development in each ASEAN country, it is suggested that the other countries in ASEAN develop a proper technology regarding GR development. As for studies in GR motivations, the 38 studies are significantly higher than for the other two criteria, making the level of evidence pertaining to this service strong. To validate the benefits of using GRs in ASEAN countries, research focusing on motivation appears to remain popular. The research on barriers is also quite dominant, with 25 articles demonstrating that the development of GRs in ASEAN countries still faces numerous obstacles.

The grouping of GR development levels is determined by a sequence of development scenarios from Stage 1 to Stage 6 based on the 70 research papers gathered. An extensive variety of papers, from analyzing urban issues to evaluating GRs, have been compiled. In terms of Singapore and Malaysia, the GR performance evaluation articles reveal that GR implementation has been accomplished and examined for quality (e.g., Azis et al. [72], Hwang [98]). However, Malaysia also reported numerous barriers (Table 9). The community's reluctance to implement GRs is a result of the government's lack of emphasis on their widespread installation.

Thailand and the Philippines occupied Stage 5 despite having few research articles on the development of GRs because the GR system has been implemented on the largest green roof farm building in Thailand. In the Philippines, a comparative analysis of the quality and prices of GR suppliers reveals that the market for GR suppliers in the Philippines is competitive. However, the disparity in the government's "Promotion" stage is still relatively small. To accelerate development in Thailand and the Philippines, government assistance such as guidebooks and regulations is necessary.

Indonesia and Vietnam are in Stage 3 of "GR Research and Development," despite the fact that Indonesia has quite a few articles containing sixteen study GRs. Indonesia continues to struggle with benefit validation despite conducting a great deal of research on the topic of motivation with nine studies, the majority of whose results were model-based. This imbalance between model and actual survey results is shown in several parameters of UHI effects (Section 3.2). In the meantime, the number of GR study articles in Vietnam remains very low. In addition to a dearth of interest in using GRs and a lack of expertise, the development of GRs in these two nations has stalled at Stage 3.

Brunei, Myanmar, Cambodia, and Laos have conducted limited research on GRs, resulting in a low level of development in these four nations. In Brunei and Myanmar, evaluation is still limited to barriers to the implementation of GRs. Laos has not conducted any research on GRs, whereas Cambodia is still evaluating their ability to combat flooding.

Compared to all challenges, the challenge in ASEAN countries is clearly reported to increase the interest of using GRs. This must also be supported by government regulations that are still minimally implemented in ASEAN nations. In some instances, there is no definitive standard for the development of GRs, so price fluctuations remain highly variable. Several GR development alternatives are explained in Section 4.2.

*4.2. Perspective and Future Needs of GR Development in ASEAN Countries*

The literature review highlights some important findings of drivers where policy pressure can increase green roof development level (this is the case reported, e.g., by Ismail [28] and Fauzi [29]). Moreover, differences in Singapore's and Malaysia's rules indicate that each nation's regulations are created according to its own needs. It is important to point out, ASEAN countries without the support of government regulations have difficulty developing green roofs. The advancement of technology is an additional factor that contributes to the development of GRs. However, through examining the reviews,

just a limited source pertaining to advanced technology was discovered. This is also a significant gap that makes it difficult for other countries outside of Singapore and Malaysia to grow in GR development.

The present investigation on factors that motivate GR implementation in ASEAN countries has been found up to ten in all ASEAN countries. Some of the aspects that can be highlighted are:

- GRs can reduce energy consumption in buildings by up to 57%, reduce roof surface temperatures when compared to bare roofs up to 23.8 °C, and reduce the room temperature to 14 °C. It should be noted that heating load reductions can be influenced by substrate [73], vegetation [76], and insulating material [81]. Some researchers describe alternative material such as lightweight concrete deck that can improve GR properties [40,51].
- GRs aid in water management, owing to their high water retention potential and increased runoff quality. The application of GRs in ASEAN countries is highly appropriate due to the region's abundant precipitation. GRs can control up to 98.8% of flood water retention. However, the retention is high only for low- to moderate-intensity rainfall. Countries with significant precipitation, such as the Philippines, are highly interested in GR water management research (e.g., as described in the case reported by Orozco [36]). The use of hydrogel to increase water retention on the substrate is an interesting innovation to be developed (from research by Asinas [57]). Other countries with heavy precipitation, such as Indonesia, are encouraged to conduct studies on the water management of GRs.
- The present review study also signals biodiversity, which is the fact that GRs can promote urban biodiversity and enhanced habitat connectivity between flora and fauna as was shown in research in Singapore [113]. A habitat is created from many environmental factors. We found no literature on biodiversity in other ASEAN nations during our search. We need further research on the effect of environment and topography on the biodiversity of the GR.

However, the study of GR barriers in ASEAN countries is particularly prominent. Of the 70 studies in the literature, 25 studies on the barriers of GRs in ASEAN countries were majorly discussed in Section 4.2. All barriers can be overcome by conducting and developing more research on the drivers. Several prospective research recommendations on the topic of drivers are as follows:

1. Investigating alternative materials that can enhance the efficacy of GRs and reduce installation costs.
2. Developing intelligent technology that can expedite the retrieval of GR data in order to accelerate the pre-development, construction, and maintenance processes.
3. Planning guidelines and economic schemes that can be referred to in the future when creating government regulations.
4. Conducting research on the benefits of GRs at each city scale (e.g., residential buildings, commercial buildings, and high-rise buildings) to educate users on the advantages of GR use.
5. At this time, numerous diverse data can be suggested for calculating the mean and median value for each parameter. Due to the heterogeneity of the parameters in the calculated studies, it was deemed statistically invalid to compute such values. Evaluation of performance can serve as an indicator for evaluating the performance of green roofs in ASEAN countries.

## 5. Conclusions

This study systematically reviewed the recent literature to identify the current development of green roofs in ASEAN countries.

According to our findings, GRs give multiple benefits to both users and the ecosystem. GRs effectiveness is influenced by many factors such as the weather and the location of ASEAN nations that is ideal for optimizing the properties of GRs.

However, a review of the literature on GR research in ASEAN countries revealed that the level of development of GRs in ASEAN countries is not evenly distributed. Most notably, the key challenges are similar, because of a lack of supporting regulations, lack of expertise in green roofs, and expensive installation prices. Regulatory concerns, in the opinion of the present authors, must be issued through the governments' technical authority so that they have the capacity and legal basis. While the regulations are being finalized, the development of green roof technology in accordance with the climate features of ASEAN countries must be prioritized by implementing the appropriate green roof technology, so the installation costs for building GRs can be decreased.

The scientific potential of the present work is that it allows us to understand the specific beneficial performance of green roofs in ASEAN countries and propose a future strategy to overcome the barriers. Increased public awareness is expected to increase the ASEAN region's GR level in the long term.

**Author Contributions:** Conceptualization, H.C.P., T.S. and A.C.; methodology, H.C.P., T.S. and A.C.; formal analysis, H.C.P.; writing—original draft preparation, H.C.P.; writing—review and editing, H.C.P., T.S. and A.C.; revisions, H.C.P., T.S. and A.C.; supervision, T.S. and A.C.; project administration, T.S. and A.C.; funding acquisition, T.S. and A.C. All authors have read and agreed to the published version of the manuscript.

**Funding:** This study is supported by research funds to extend knowledge from OROG Thailand Scholarship, Sustainable Innovative and Energy Efficient Construction Material (SIE-CON); and Infinity Concrete Technology, Co., Ltd.

**Institutional Review Board Statement:** Not applicable.

**Informed Consent Statement:** Not applicable.

**Data Availability Statement:** Not applicable.

**Acknowledgments:** The authors are grateful to the Suranaree University of Technology; Sustainable Innovative and Energy Efficient Construction Material (SIE-CON); and Infinity Concrete Technology, Co., Ltd. for providing support financially and for allowing us to use its facilities.

**Conflicts of Interest:** The authors declare no conflict of interest.

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
