# Peer review of "Green Roof Development in ASEAN Countries: The Challenges and Perspectives"

_sustainability, doi:10.3390/su15097714_

Round 1

Reviewer 1 Report

This article reviews the GR related articles through several major databased and MDPI. The review process is carried extensively and with details explanation. However, the findings and conclusion still are not insightful. For example,  "According to our findings, ASEAN society is still unaware about GRs. However, if GRs have an affordable and visually acceptable design, they have the potential for social acceptability. Furthermore, user-specific experiences and expectations are essential to consider. To expand their awareness, there is a need to directly introduce the benefits of GRs to society." in the result session did not provide any insightful result. This ends up the research conducted is not so meaningful except the hardworking for collecting the data.

Authors are advised to enhance the session 3 and session 4 extensively to be justified as a research paper.

Author Response

Dear reviewer,

Point 1 :
However, the findings and conclusion still are not insightful. Authors are advised to enhance the session 3 and session 4 extensively to be justified as a research paper.

Response 1:

Thank you. In response to the comment, we have added session 3 and sorted the data for more focused results. we also added session 4 to be 2 discussions.
4.1. Challenges of GRs development in ASEAN
4.2. Perspective and future needs of GRs development in ASEAN

Thanks so much for the valuable comments

Reviewer 2 Report

This is a review paper which investigates green roofs in ASEAN. The studies promotos more use of Green roofs to conduct more research on the drivers aspects, comprising supporting regulations and advance innovative technology to enhance. The paper critise the ASEAN about grean roofs.
The authors utilized more than 100 papers. Therefore, it collects a general idea about  green roofs and critise literaure

I have a few comments.

Include more photos and figures
Conclusion section section should be imprvode. Give more results with bullets.
Add section for lack of current studies
Add section for future needs

Author Response

Dear reviewer,

Point 1 :
Include more photos and figures

Response 1 :
Thank you for your comment. I add some pictures of the experimental GRs design in several results parameters to describe how the selected papers conduct the experiment.

Point 2:
Conclusion section section should be imprvode.
Give more results with bullets.

Response 2 :
Thank you, I added a conclusion item to be 2 parts.
4.1. Challenges of GRs development in ASEAN

4.2. Perspective and future needs of GRs development in ASEAN

Point 3:
Add section for lack of current studies
Add section for future needs

Response 3:
Thank you. I discuss the lack of current studies in section 4.1
and also I added 5 possible future needs in section 4.2

Thanks so much for the valuable comments

Reviewer 3 Report

1- Some results should be added to the abstract.

2- The authors collected data from 70 articles (after justifications), they only presented the findings. Where is the statistical analysis of the findings?

3- The conclusion may be more precise and indicate, in points, the main outcomes of the study.

4- Some of the tables should be better represented graphically, instead.

5- Some typos errors should be corrected. Please, revise the whole manuscript.

Author Response

Dear reviewer,

Point 1:
Some results should be added to the abstract.

Response 1:
Thank you, we added the performance of each parameter of the result to the abstract. "The review recommends promoting the use of GRs, which have the potential to reduce energy consumption by up to fifty percent, reduce the outdoor surface temperature up to 23.8 °C, and reduce the room temperature to 14 °C. The use of GRs can also mitigate runoff control by up to 98.8% to avoid the risk of flooding in ASEAN which has high rainfall. "

Point 2:
The authors collected data from 70 articles (after justifications), they only presented the findings. Where is the statistical analysis of the findings?

Response 2:
Thank you. In this review, we are not provided the statistical analysis. we used the systematic literature review (SLR) in terms of collecting the data. Here is I attached the PRISMA Flow Diagram. 

Point 3:
The conclusion may be more precise and indicate, in points, the main outcomes of the study.

Response 3:
Thank you. The main outcomes are to investigate the level of development GRs in ASEAN countries and provide qualified support that can help researchers in designing urgent GRs for future research in ASEAN. The outcomes resulted in the status of GRs level development in (section 3.2. Green Roofs Stage in ASEAN), and the future needs recommendation is provided in (section 4.2. Perspective and future needs of GRs development in ASEAN)

Point 4:
Some of the tables should be better represented graphically, instead.

Response 4:
Thank you. we have changed several tables to be graphics and sorted several data on the table to be easier to understand. 

Point 5:
Some typos errors should be corrected. Please, revise the whole manuscript

Response 5:
Thank you, we have checked some typos and revised it. However, we will send the manuscript with English language proof if necessary.

Thanks so much for the valuable comments 

Round 2

Reviewer 1 Report

There are some minor spelling errors. Other than that the content has been improved according to my previous comments. 

Author Response

Point 1 :
There are some minor spelling errors. Other than that the content has been improved according to my previous comments. 

Response 1:

Thank you for your comment. We have rechecked the misspelling and English grammar again.

Thanks so much for the valuable comments 
